

# Cognitive-behavioral treatment for insomnia and mindfulness-based stress reduction in nurses with insomnia: a non-inferiority internet delivered randomized controlled trial

Wanran Guo[1], Nabi Nazari[2] and Masoud Sadeghi[2]

[1] School of Public Policy and Administration, Nanchang University, Nanchang, Nanchang, China
[2] Department of Psychology, Faculty of Human Sciences, University of Lorestan, Khorramabad, Lorestan, Iran

## ABSTRACT

**Background**. Insomnia is a highly prevalent sleep disorder frequently comorbid with mental health conditions in nurses. Despite the effectiveness of evidence-based cognitive behavioral therapy for insomnia (CBT-I), there is a critical need for alternative approaches. This study investigated whether internet-delivered mindfulness-based stress reduction (IMBSR) for insomnia could be an alternative to internet-delivered CBT-I (ICBT-I).

**Objective**. The hypothesis was that the IMBSR would be noninferior to the ICBT-I in reducing the severity of insomnia among nurses with insomnia. Additionally, it was expected that ICBT-I would produce a greater reduction in the severity of insomnia and depression than IMBSR.

**Method**. Among 240 screened nurses, 134 with insomnia were randomly allocated (IMBSR, $n = 67$; ICBT-I, $n = 67$). The assessment protocol comprised clinical interviews and self-reported outcome measures, including the Insomnia Severity Index (ISI), Patient Health Questionnaire-9 (PHQ-9), the 15-item Five Facet Mindfulness Questionnaire (FFMQ), and the Client Satisfaction Questionnaire (CSQ-I).

**Results**. The retention rate was 55% with 77.6% ($n = 104$) of participants completing the study. At post-intervention, the noninferiority analysis of the ISI score showed that the upper limit of the 95% confidence interval was 4.88 ($P = 0.46$), surpassing the pre-specified noninferiority margin of 4 points. Analysis of covariance revealed that the ICBT-I group had significantly lower ISI (Cohen's $d = 1.37$) and PHQ-9 (Cohen's $d = 0.71$) scores than did the IMBSR group. In contrast, the IMBSR group showed a statistically significant increase in the FFMQ-15 score (Cohen's d = 0.67). Within-group differences showed that both the IMBSR and ICBT-I were effective at reducing insomnia severity and depression severity and improving mindfulness.

**Conclusion**. Overall, nurses demonstrated high levels of satisfaction and adherence to both interventions. The IMBSR significantly reduced insomnia severity and depression, but the findings of this study do not provide strong evidence that the IMBSR is at least as effective as the ICBT-I in reducing insomnia symptoms among nurses with insomnia. The ICBT-I was found to be significantly superior to the IMBSR in reducing insomnia severity, making it a recommended treatment option for nurses with insomnia.

Corresponding author
Nabi Nazari, nazariirani@gmail.com

## BACKGROUND

Insomnia, characterized by difficulty falling asleep or staying asleep (*American Psychiatric Association, 2022*), has become prevalent among nurses during the coronavirus disease 2019 (COVID-19) pandemic (*Pappa et al., 2021*; *Xia et al., 2021*). Insomnia adversely impacts quality of life (*Lu et al., 2021*), patient care quality (*Huo et al., 2022*), work performance (*Hui & Grandner, 2015*), immune system functionality (*De Almondes, Marín Agudelo & Jiménez-Correa, 2021*), the occurrence of medical errors, the experience of persistent fatigue, and burnout in nurses (*Knutson, 2018*; *Hu et al., 2020*). Research has shown a bidirectional causal relationship between insomnia and a range of mental health conditions, including anxiety (*Alvaro, Roberts & Harris, 2013*), depression (*Waage et al., 2014*; *Meaklim et al., 2021*), and emotion dysregulation (*Kahn, Sheppes & Sadeh, 2013*). Despite its prevalence and clinical relevance, insomnia among healthcare workers (HCWs) is often underdiagnosed and undertreated (*Ogeil et al., 2020*). Furthermore, medication is a typical treatment for insomnia, but it lacks long-term efficacy and is associated with risks of unwanted side effects, such as dependency, withdrawal, and headaches (*Krystal & Lichstein, 2016*). To address these limitations, nonpharmacological interventions have gained prominence as the preferred treatment for insomnia.

Cognitive behavioral therapy for insomnia (CBT-I) is the first-line nonpharmacological treatment for insomnia, offering significant benefits with minimal risks (*Taylor, Lichstein & Morin, 2014*; *Rios et al., 2019*) and longer-lasting effects compared to medication (*Mitchell et al., 2012*). As a multicomponent intervention, CBT-I is also effective in addressing depressive symptoms that frequently co-occur with insomnia (*Qaseem et al., 2016*; *Sateia et al., 2017*). CBT-I targets multiple mechanisms known to perpetuate insomnia. For example, CBT-I promotes sleep drive consolidation through consistent sleep schedules, identifies and modifies maladaptive behaviors and cognitions related to sleep, and addresses counterproductive attempts to force sleep *via* sleep restriction therapy. Additionally, CBT-I allows individuals to use relaxation techniques to manage sleep-related anxiety, ultimately strengthening bed-sleep associations and promoting a calm presleep state (*Baglioni et al., 2020*).

Although CBT-I is indeed an effective approach for managing insomnia, CBT-I may not offer a comprehensive solution to all the contributing factors underlying this sleep disorder. Specifically, CBT-I may not adequately address transdiagnostic mechanisms such as elevated cognitive arousal and emotion dysregulation, which are closely linked to sleep disturbances and instances of treatment-resistant insomnia (*Palagini et al., 2018*; *Oldsen, Smith & Brown, 2014*; *Ellis et al., 2021*). Moreover, CBT-I may not be accessible to all patients because of the limited availability of qualified therapists, low treatment
acceptance, and poor adherence (*Lovato et al., 2014*; *Koffel, Bramoweth & Ulmer, 2018*). Even after CBT-I treatment, half of the patients may still experience insomnia symptoms, even if they meet the criteria for remission (*Lu et al., 2021*). Certain core components of CBT-I, such as sleep restriction, can induce side effects such as fatigue and extreme sleepiness, which can negatively impact treatment adherence and potentially restrict the effectiveness of the intervention (*Riedel & Lichstein, 2001*; *Stevens, 2015*). Therefore, the development of novel and innovative approaches for insomnia treatment may necessitate a focus on transdiagnostic mechanisms and increased accessibility. Mindfulness-based interventions (MBIs) offer a solution by incorporating mindfulness techniques, a practice complementary to CBT-I (*Garland et al., 2016*; *Ong, Ulmer & Manber, 2012*; *Palagini et al., 2014*).

Central to MBIs is the practice of mindfulness, defined as the nonjudgmental observation of present-moment experiences, encompassing both internal (thoughts, emotions) and external (sensory details) stimuli (*Cayoun, Francis & Shires, 2019*; *Gupta, 2022*). This practice cultivates a deliberate focus on the present with acceptance, openness, and compassion. Insomnia is linked to negative thought patterns. MBIs target these patterns by promoting a shift from automatic reactions to a more detached, observer-like position toward thoughts and emotions (*Segal, Williams & Teasdale, 2002*). This approach, termed cognitive defusion, effectively disrupts dysfunctional thinking patterns observed in various mental health conditions, suggesting its potential for insomnia interventions. By shifting from outcome-driven thinking to a process-oriented observation of thoughts and feelings, these interventions can alleviate emotional distress, particularly during stressful events (*Davis & Hayes, 2011*; *Hofmann et al., 2010*).

Mindfulness meditation and yoga, the core components of MBIs, induce changes in brain regions associated with emotion regulation, attention regulation, and self-referential processing (*Hölzel et al., 2011*; *Pascoe, Thompson & Ski, 2017*). By focusing on the present moment and bodily sensations rather than intrusive thoughts, MBIs promote relaxation and prepare the body for sleep (*Rusch et al., 2019*). Yoga complements mindfulness by integrating intentional movements and controlled breathing techniques, further facilitating relaxation. Beyond formal sessions, integrating mindfulness into daily routines enhances awareness and presence, leading to improvements in sleep quality and overall well-being. MBIs target transdiagnostic mechanisms, such as stress responses, emotion dysregulation, and rumination, which play crucial roles in the development and maintenance of insomnia. In this context, the key advantages of MBIs are their potential to directly reduce presleep arousal, promote emotion regulation skills (*Ma et al., 2018*), and help patients avoid the short-term unpleasant consequences of CBT-I (*e.g.*, sleep restriction and exposure therapy; *Cincotta et al., 2010*).

Stress can disrupt sleep initiation (*i.e.*, increased levels of the hormone cortisol), maintenance (*i.e.*, increased cognitive arousal; *Kalmbach, Anderson & Drake, 2018*; *Slavish et al., 2018*), and consolidation (*i.e.*, restorative stages of sleep; *Liu et al., 2019*). Individuals with insomnia frequently report heightened psychological distress, difficulty coping with stress, and increased presleep arousal (*Alimoradi et al., 2021*; *Vandekerckhove & Wang, 2017*). Therefore, mitigating stress has the potential to enhance both the quality

and duration of sleep. In this context, mindfulness-based stress reduction (MBSR) has emerged as a promising approach for tackling insomnia and sleep disturbances (*Black et al., 2015*; *Camino et al., 2022*; *Kim et al., 2022*; *Ong, Ulmer & Manber, 2012*; *Chen et al., 2020*). Frontline medical staff experiencing high stress levels are at greater risk of sleep disorders and PTSD-related symptoms (*Liu et al., 2022*; *Yin et al., 2020*). MBSRs help nurses cope with stress and maintain their overall well-being and job satisfaction (*Lomas et al., 2018*; *Lamothe et al., 2016*; *Bohlmeijer et al., 2010*). Furthermore, mindfulness has been shown to be effective at reducing psychological distress, depressive symptoms, burnout, maladaptive behaviors, and negative emotions (*Lim et al., 2021*; *Reangsing, Rittiwong & Schneider, 2021*; *Li & Bressington, 2019*; *Kriakous et al., 2021*; *Fang et al., 2019*; *McClintock, Rodriguez & Zerubavel, 2019*; *Şahin, Arıcımath Özcan & Arslan Babal, 2020*). Consequently, MBSR can be an invaluable tool for nurses to manage stress, improve sleep quality, and maintain overall well-being (*Hoedl et al., 2023*).

Digital health interventions offer a practical solution to overcome accessibility barriers for insomnia treatments (*Lattie, Stil es-Shields & Graham, 2022*). Internet-delivered interventions (IDIs) provide a cost-effective and scalable approach that can reach a wide range of individuals (*Zachariae et al., 2016*). These interventions are not only affordable but also offer remote access, which is convenient for healthcare workers with demanding schedules. Moreover, the anonymity of these platforms may appeal to healthcare workers seeking privacy. Specifically, internet-delivered CBT for insomnia (ICBT-I) has been shown to be as effective as traditional CBT while potentially offering greater convenience and reach (*Riemann & Espie, 2009*; *Ritterband et al., 2009*; *Luo et al., 2020*). Additionally, internet-delivered mindfulness-based stress reduction (IMBSR) has demonstrated potential for improving sleep quality and overall well-being (*Kang et al., 2021*; *Pan et al., 2022*).

## The current study

Noninferiority randomized controlled trials (RCTs) are instrumental in appraising the effectiveness of emerging treatments in compared with established benchmarks (*Piaggio et al., 2012*). In this pivotal study, a noninferiority design plays a crucial role in meticulously evaluating emerging treatments, specifically focusing on standardized internet-delivered interventions such as the IMBSR, and comparing them against established benchmarks. A large body of evidence highlights the deleterious psychological impacts of the COVID-19 pandemic on frontline HCWs, and sleep-focused treatments are warranted to support their well-being (*De Kock et al., 2021*; *Salari et al., 2020*). Therefore, this study investigated the efficacy of internet-delivered CBT-I and MBSR for clinical nurses with insomnia.

The primary goal of this study was to assess whether internet-delivered MBSR for insomnia yields results that are at least as effective as those of ICBT-I in reducing insomnia among clinical nurses. The hypothesis was that the IMBSR would be noninferior to the ICBT-I by significantly reducing insomnia symptoms. The second aim of the study was to investigate the efficacy of internet-based formats for both MBSR and CBT-I in treating insomnia among nurses. It was hypothesized that both the IMBSR and ICBT-I would lead to significant improvements in insomnia severity, mindfulness, and depression following
the intervention. Additionally, it was expected that the ICBT-I would lead to a greater decrease in the severity of insomnia and depression than the IMBSR.

## METHODS

### Trial setting

The study was a two-armed, assessor-blinded, internet-delivered RCT with a parallel group noninferiority design and an equal allocation ratio (1:1) that compared the experimental intervention (IMBSR) with the standard treatment (ICBT-I). Ethical approval was granted from the Research Ethics Committee of Lorestan University of Medical Sciences (Approval No: IR.LUMS.REC.1399.269). The trial was prospectively registered at ISRCTN (ID: ISRCTN36198096) (*Li, Nazari & Sadeghi, 2022*). The trial spanned from June 2022 to May 2023.

The CONSORT (Consolidated Standards of Reporting Trials) checklist for noninferiority RCTs was followed to report this trial (*Piaggio et al., 2012*). Of the 240 clinical nurses screened for eligibility, 106 were excluded for various reasons, including 34 who declined to participate, 30 who had scheduling conflicts, 24 who did not meet a minimum cutoff score for clinical insomnia (15 points), and 18 who were found ineligible following clinical interviews. A total of 134 clinical nurses who met the eligibility criteria consented to participate in the study. The CONSORT diagram is presented in Fig. 1.

*The inclusion criteria were as follows:* (a) provided signed informed consent, (b) were actively employed as clinical nurses on the front lines of the COVID-19 pandemic, (c) were at least 18 years of age, and (d) internet access. The participants (e) received a formal diagnosis of insomnia based on the Structured Clinical Interview for DSM-5 Sleep Disorders (SCID-5-SD), (f) reported clinically significant moderate insomnia (15–21) to severe insomnia (22–28) symptoms on the Insomnia Severity Index (ISI) by scoring 15 points or more (*Morin et al., 2009*), and (g) demonstrated a willingness to participate. *The exclusion criteria* were as follows: (a) diagnosed with COVID-19, (b) received psychological treatment for insomnia within the last year, (c) missed three or more scheduled sessions, and (d) currently using complex or unstable sleep medication.

### Procedure

Participants were recruited from June to December 2022 through online announcements, social media posts, public health forums, and word-of-mouth referrals. This multifaceted approach enabled the study to reach a diverse pool of potential participants, encompassing nurses from various workplaces with varying levels of experience and who represented diverse backgrounds. Additionally, nurses were encouraged to share information about the study with their colleagues, further broadening the participant base.

Eligible individuals could apply to participate in the study *via* email or phone. Participants received comprehensive information about the study's objectives, potential benefits, associated risks, and expected duration. Additionally, participants were assured of the confidentiality and anonymity of their personal information, with the flexibility to withdraw their consent at any time during the study. Potential participants were invited
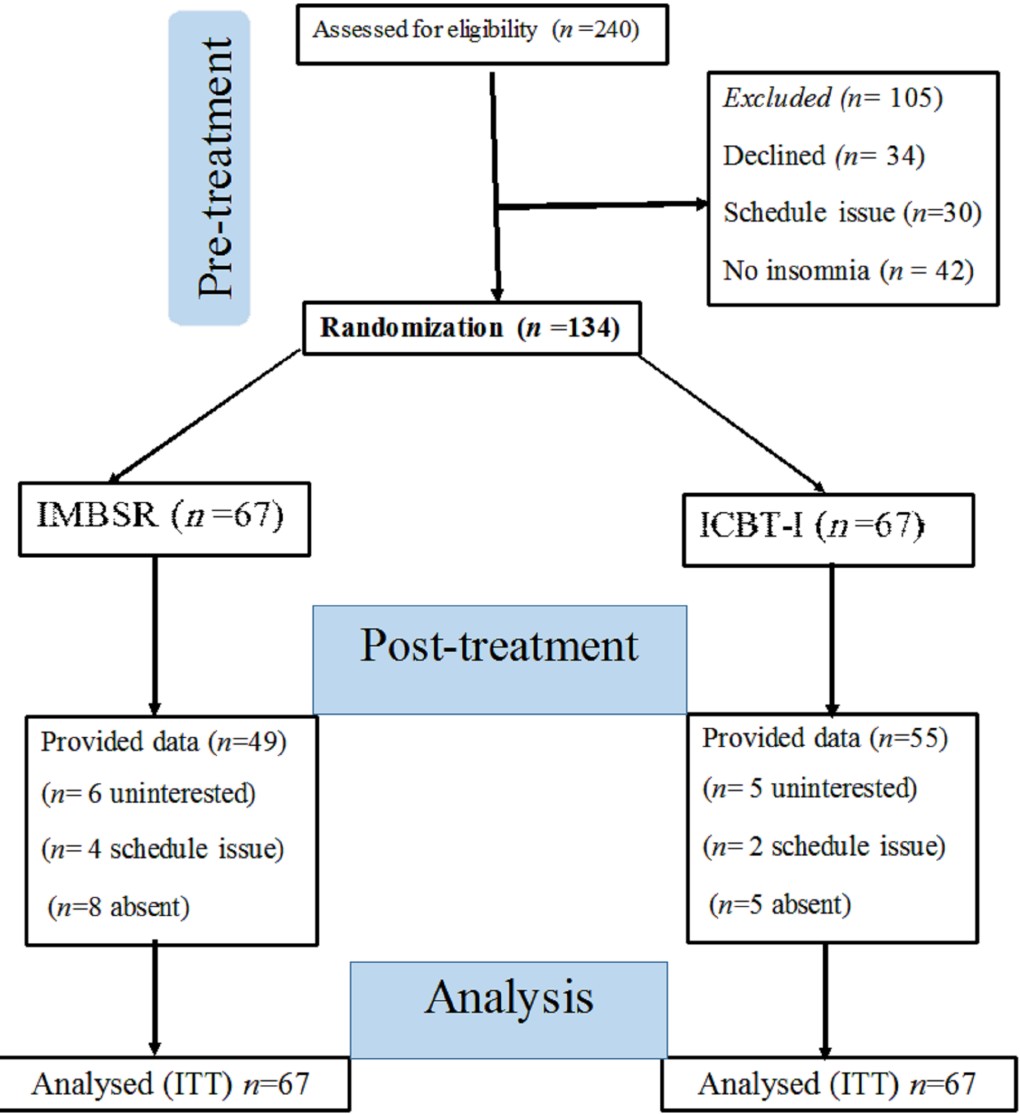

**Figure 1** **The participants flow chart diagram.** Note: IMBSR, internet-delivered mindfulness-based stress reduction; ICBT-I, internet-based cognitive behavioural therapy for insomnia; ITT, intent-to-treat.

to provide an email address to receive an invitation letter containing both the informed consent form and the primary outcome measure.

Only nurses who provided informed consent and had an insomnia severity index (ISI) score of 15 or higher progressed to the Structured Clinical Interview for Sleep Disorders (SCID). This in depth, one-hour online clinical interview was conducted by two clinical psychologists to rigorously assess participants against the eligibility criteria. This comprehensive screening process ensured that only individuals with clinically significant insomnia were included in the study, minimizing potential risks to participants and enhancing the validity of the study's findings. Finally, eligible participants were randomized to receive either the IMBSR or ICBT-I intervention.

## Interventions

Participants were requested to maintain their regular lifestyle throughout the study, including daily activities, exercise regimen, dietary habits, and medication routines. Additionally, any individuals undergoing psychological interventions needed to disclose such treatments. Both the IMBSR and ICBT-I commenced with an initial motivational interview module designed to motivate participants to change and improve their adherence.

## The IMBSR program

The IMBSR program was administered in eight 2-hour weekly sessions (*Kabat-Zinn, 2013*; *Teasdale, Segal & Williams , 2014*), plus one 6-hour weekend intensive silent retreat. The course aimed to foster participants' mindfulness and encourage them to respond adaptively to internal and external stressors. The core components of the program included (a) motivational interview, (b) theoretical underpinnings of mindfulness within the mind-body context to learn mindful awareness of the body, (c) practice of three formal mindfulness techniques: hatha yoga (followed by body scan), sitting meditation, and optional meditation, (d) cultivating mindful responses to stress and growing the ability to respond mindfully and deliberately to stress rather than falling into preconditioned, habitual, and reactive responses, and (e) plans for future practice and maintaining treatment gains, and (f) daily mindfulness practices and maintaining gains: participants received weekly homework assignments that reinforced the learned concepts. These may include practicing guided meditations at home (20–30 min), completing mindful movement exercises such as mindful yoga or body scans, and integrating mindful sleep hygiene practices into bedtime routines (*e.g.*, mindful breathing exercises before bed) to promote a more sleep-conducive environment.

## The ICBT-I program

The ICBT-I program included eight 2-hour weekly sessions (*Edinger & Carney, 2015*; *Perlis & Al, 2008*). The goal of CBT-I is to help individuals develop healthy sleep habits and overcome insomnia by addressing both behavioral and cognitive factors that contribute to sleep difficulties. The program consisted of (a) motivational interview, (b) sleep hygiene module, (c) stimulus control, (d) sleep restriction, (e) relaxation techniques (*i.e.,* muscle relaxation, breathing exercise), (f) cognitive therapy (*i.e.,* cognitive restructuring), and (g) preventing relapse (see Appendix 1 for a detailed description of interventions).

## Outcomes

The outcome measures included the ISI, the nine-item Patient Health Questionnaire (PHQ-9; *Kroenke, Spitzer & Williams, 2021*), the 15-item Five Facet Mindfulness Questionnaire (FFMQ-15; *Gu et al., 2016*; *Khanjani et al., 2022*), and the Client Satisfaction Questionnaire-Internet (CSQ-I; *Boßet al., 2016*). The participants completed the self-reported instruments at Time 1 (baseline to allocation) and Time 2 (postintervention), with reminder emails sent within one week at each time point. The study timeline is presented in Table 1.

**Table 1  Study schedule of enrolment, intervention, and assessment.**

| | Study Period | | | | |
|---|---|---|---|---|---|
| **Item** | **Enrolment** | **Allocation** | **Post-allocation** | | |
| **Timepoint** | *Time1* | *t0* | *t1* | *t8* | *Time2* *Immediately after intervention* |
| **Enrolment** | | | | | |
| Eligibility screen | × | | | | |
| Informed consent | × | | | | |
| Allocation | | × | | | |
| **Interventions**: | | | | | |
| *Group A: IMBSR* | | | | | |
| *Group B: ICBT-I* | | | | | |
| **Assessments**: | | | | | |
| Sociodemographic | × | | | | |
| Clinical interview | × | | | | × |
| Insomnia severity | × | | | | × |
| Depression | × | | | | × |
| Mindfulness | × | | | | × |
| Client satisfaction | | | | | × |

Notes.
ICBT-I, Internet-delivered Cognitive-behavioural Therapy for Insomnia; IMBSR, Internet-delivered Mindfulness-Based Stress Reduction.

## Primary outcome
### Insomnia severity
The ISI was used to assess insomnia severity in nurses over the preceding two weeks. The ISI is a 7-item self-report questionnaire with a 5-point Likert scale, with higher scores indicating more severe insomnia. A score of 14 or lower indicates subthreshold insomnia, while a score of 15 or higher indicates clinical insomnia. The ISI has good internal consistency (Cronbach's $\alpha = 0.84$).

## Secondary outcome
### Mindfulness
The FFMQ-15 is a self-report questionnaire that measures five dimensions of mindfulness: observing, describing, acting with awareness, nonjudgment, and nonreactivity. Higher scores on the FFMQ-15 indicate higher levels of mindfulness. The FFMQ-15 is a reliable and valid measure of mindfulness, with good internal consistency (Cronbach's $\alpha = 0.84$).

### Depression
The PHQ-9 is a nine-item self-report questionnaire that measures the severity of depression over the past two weeks. Items are rated on a 4-point Likert scale, with higher scores indicating greater depression severity. The total score ranges from 0 to 27, with scores of 5 or higher suggesting clinical depression. The PHQ-9 has good internal consistency (Cronbach's $\alpha = 0.85$).

*Client satisfaction*

The CSQ-I is a self-reported measure of client satisfaction with online interventions. Eight items were rated on a four-point Likert scale, with higher scores on the CSQ-I indicating greater satisfaction with the online interventions received. The CSQ-I is a reliable measure of client satisfaction (Cronbach's $\alpha = 0.88$).

## Sample size

Sample size determination was meticulously conducted to adhere to stringent methodological standards, guided by the recommended minimal important difference (MID) for noninferiority trial designs, as elucidated by *Hwang & Morikawa (1999)*. We set the noninferiority margin at half of the MID, equivalent to an eight-point reduction on the Insomnia Severity Index (ISI), based on the seminal work of *Morin et al. (2011)*. With a significance level ($\alpha$) of 0.05 and a power level of 80%, our power analysis was conducted meticulously to ensure robustness. Accounting for a projected 25% dropout rate, our calculated sample size of 134 participants was determined with precision to guarantee adequate statistical power.

## Randomization

A permuted block randomization schedule was generated using a computerized random number generator and a random sequence of block sizes (*i.e.,* 4, 6, and 8). The randomization schedule was stored in a secure location, and only authorized personnel had access to it. The independent investigator was responsible for assigning participants to the study arms. The randomization schedule was stored securely, and only authorized personnel had access to it. The treatment allocation was concealed from both the researchers and the participants until all posttreatment assessments had been completed.

## Harms

Although CBT-I and MBSR are generally safe treatments for insomnia, the potential for adverse events and harm were systematically monitored throughout the trial. Participants who exhibited heightened psychological distress, temporary sleep disruption, worsening insomnia, depersonalization, or other concerns were offered support by licensed clinical psychologists and general practitioners through both email and telephone consultations. The clinical team underwent comprehensive training in psychological assessments, structured interviews, and ethical considerations in clinical research.

## Statistical analyses

All statistical analyses were conducted using the SPSS software (version 25; IBM Corp., Armonk, NY, USA). Two-tailed tests were employed, with a significance level (alpha) set at 0.05, except for the noninferiority analysis, which utilized a one-tailed test. Adopting an intent-to-treat (ITT) approach, all initially randomized participants were included, irrespective of withdrawal or dropout. Missing data were handled through imputation using the last observation carried forward (LOCF) method, substituting missing data points with the most recent recorded values. Descriptive statistics were utilized to calculate demographic characteristics and outcome measures. Categorical variables were analyzed

using chi-square tests, while continuous variables were examined using with independent t tests. Program feasibility and acceptability were assessed through the recruitment rate, retention rate, and client satisfaction using the CSQ-I.

In the context of a noninferiority trial, the F-statistic test and the corresponding $p$-value were applied to assess the variance between the two treatment groups. This step is pivotal because differing variances can influence the power of the noninferiority test. A 95% confidence interval (CI) approach was employed to test the noninferiority of the IMBSR in compared with the ICBT-I, with a noninferiority margin set at 4 points on the ISI score (*Mascha & Sessler, 2011*). The $p$ value estimates the likelihood that the mean ISI score for IMBSR is significantly smaller than the mean for ICBT-I plus the noninferiority margin of 4.0, assuming that the null hypothesis is valid. In this instance, the null hypothesis posits that the experimental intervention (IMBSR) is inferior to the standard treatment (ICBT-I) by a margin equal to or greater than 4 points on the ISI. Noninferiority was confirmed if the upper bound of the one-tailed 95% CI of the mean difference between the groups on ISI score was less than 4 points in favor of the ICBT-I.

To compare the efficacy of ICBT-I with that of IMBSR, univariate analysis of covariance (ANCOVA) was performed on the ISI, PHQ-9, and FFMQ-15, with Time1 data serving as a covariate to mitigate bias (*i.e.,* preexisting group disparities). The assumptions, including a normal distribution, equal error variances, and homoscedasticity, were verified before conducting the ANCOVA. Treatment effects are presented in terms of Cohen's $d$ and partial eta-squared ($\eta^2 p$) values. Paired t tests were computed to evaluate within-group changes between Time 1 and Time 2. The number of participants exhibiting subthreshold insomnia at Time 2 was also reported.

## RESULTS

### Descriptive characteristics
The descriptive characteristics of the sample are presented in Table 2. At baseline, t tests revealed no significant differences in continuous variables, indicating successful randomization. The study included 134 clinical nurses aged 25 to 48 years ($M = 33.12$ years; SD = 5.53). This sample population had a significant association with marital status ($\chi^2 = 9.67$, $p = 0.002$), with a larger proportion being in relationships (64.9%, $n = 87$) than single (35.1%, $n = 47$). The retention rate was 55%, with 77.6% ($n = 104$) of participants completing the study. At Time 2, 82% ($n = 55$) of the ICBT-I participants and 73% ($n = 49$) of the IMBSR participants completed the programs. The CSQ-I showed high satisfaction with the ICBT-I ($M = 26.37$, $SD = 4.87$) and the IMBSR ($M = 27.69$, $SD = 3.99$).

### Treatment results
According to the pretreatment data, there was no significant difference in variance between the two treatment groups (F-statistic = 0.28, $P = .59$). At Time 2, the estimated marginal mean ISI score for the IMBSR ($M = 14.99$) exceeded that of the ICBT-I ($M = 10.93$) by 4.06 points. The upper limit of the one-tailed 95% CI for the mean difference is 4.88, which was greater than the non-inferiority margin of 4. The $p$-value of 0.46 indicates that there

**Table 2  Demographic characteristics of the sample, $n = 134$.**

| Item characteristic | Baseline | IMBSR group | ICBT-I group | Test | $p$ value |
|---|---|---|---|---|---|
| **Categorical variables** | | | | | |
| Sex, $n$ (%) | | | | | |
|    Women | 84 (62.7) | 41 | 43 | $\chi^2 = 8.62$ | 0.003 |
|    Man | 50 (37.3) | 26 | 24 | | |
| Marital status, $n$ (%) | | | | | |
|    Single | 47 (35.1) | | | $\chi^2 = 9.67$ | 0.002 |
|    In relationship | 87 (64.9) | | | | |
| **Continues variables, mean (SD)** | | | | | |
| Age, year | 33.12 (5.53) | 33.44 (5.54) | 32.79 (5.59) | $t(132) = 0.68$ | 0.49 |
| Insomnia | 16.67 (1.44) | | | $t(132) = -0.41$ | 0.68 |
| Depression | 8.82 (2.26) | | | $t(132) = -0.15$ | 0.88 |
| Mindfulness | 32.51 (4.74) | | | $t(132) = -1.01$ | 0.32 |

**Notes.**

$n$, frequency; M, mean; SD, standard deviation; t, independent $t$ test to compare groups; ICBT-I, Internet-delivered Cognitive-behavioural Therapy for Insomnia; IMBSR, Internet-delivered Mindfulness-Based Stress Reduction.

is not sufficient evidence to reject the null hypothesis of inferiority, indicating that the IMBSR did not prove to be noninferior to the ICBT-I in reducing ISI scores at Time 2.

To ensure the robustness of the analysis, no violations of ANCOVA assumptions were identified ($P > .05$). The ICBT-I demonstrated *significantly* lower insomnia severity and depression scores than did the IMBSR. An ANCOVA was conducted to assess the effect of the intervention on insomnia severity, controlling for baseline scores. The analysis revealed a significant main effect of the intervention, $F(1, 131) = 51.88$, $p < .001$, $\eta^2 p = .28$. The effect size was large, indicating a substantial impact of the intervention on insomnia severity. The ANCOVA results highlighted a substantial decrease in the ISI score ($F(1, 131) = 61.52$, $P < 0.001$, $\eta^2 p = 0.32$, *Cohen's d* $= 1.30$, 95% CI [0.92–1.68]) and a lower PHQ-9 score ($F(1, 131) = 28.65$, $P < 0.001$, $\eta^2 p = 0.14$, Cohen's $d = 0.71$, 95% CI [0.34–1.08]) for the ICBT-I. Conversely, the IMBSR participants reported higher mindfulness scores at Time 2 ($F(1, 131) = 20.14$, $P < 0.001$, $\eta^2 p = 0.13$, Cohen's $d = -0.67$, 95% CI [−1.01 to −0.32]).

Within-group differences showed that both the IMBSR and ICBT-I were effective at reducing insomnia severity and depression severity and improving mindfulness. At Time 2, the paired $t$ test results showed that IMBSR led to significant reductions in insomnia severity ($t(66) = 7.64$, $p < .001$) and depression severity ($t(66) = 5.15$, $p < .001$) and improvements in mindfulness ($t(66) = 8.32$, $p < .001$). At Time 2, the paired $t$ test results showed that ICBT-I led to significant reductions in insomnia severity ($t(66) = 13.58$, $p < .001$) and depression severity ($t(66) = 11.56$, $p < .001$) and improvements in mindfulness ($t(66) = 2.32$, $p = .02$). Additionally, a total of 83 participants (62.0% of the sample, $n = 134$) reported ISI scores lower than 15, indicating no moderate clinical insomnia. Within the treatment groups, there were 30 participants (44.8%) in the IMBSR group and 53 participants (79.1%) in the ICBT-I group. These findings suggest that a significantly greater proportion of participants in the ICBT-I group achieved scores

| Table 3 | Means and standard deviations. | | | |
|---|---|---|---|---|
| **Item** | **IMBSR** | | **ICBT-I** | |
| | Time 1 | Time2 | Time 1 | Time2 |
| Insomnia severity M(SD) | 17.16 (1.79) | 14.99(2.69) | 16.71 (1.64) | 10.93(3.01)) |
| Depression M(SD) | 8.79 (2.46) | 7.58 (2.20) | 8.85 (2.05) | 6.00 (2.22) |
| Mindfulness M(SD) | 32.10 (4.43) | 37.85 (4.96) | 32.93 (5.04) | 34.48 (5.15) |

**Notes.**

n, frequency; M, mean; SD, standard deviation; ICBT-I, Internet-delivered Cognitive-behavioural Therapy for Insomnia; IMBSR, Internet-delivered Mindfulness-Based Stress Reduction.

indicative of no moderate clinical insomnia than did those in the IMBSR group. The means and standard deviations for the treatment outcomes are presented in Table 3.

# DISCUSSION

Nurses faced heightened risks of poor sleep and insomnia during the COVID-19 pandemic, impacting their overall well-being (*Zhang et al., 2020*). The primary hypothesis was that the IMBSR would demonstrate noninferiority to the ICBT-I in reducing insomnia symptoms, with both interventions hypothesized to induce significant changes in insomnia severity, mindfulness, and depression. While both the IMBSR and ICBT-I led to notable improvements, the study did not support the hypothesis that the IMBSR is noninferior; instead, the ICBT-I produced a more substantial reduction in insomnia severity and depression. Moving forward, the study emphasizes the need for nuanced interventions tailored to the unique challenges faced by nurses during times of heightened stress, such as the COVID-19 pandemic. Additionally, ICBT-I produced a more substantial reduction in insomnia severity and depression.

These findings cannot provide strong evidence that the IMBSR is at least as effective as the ICBT-I in reducing insomnia symptoms among nurses with insomnia. Consistent with a previous noninferiority RCT, the IMBSR was less effective than the ICBT-I for reducing insomnia symptoms, immediately at post-intervention (*Garland et al., 2014*). Both the IMBSR and ICBT-I demonstrated significant changes in insomnia severity and depression postintervention. These findings are consistent with previous studies that have supported the effectiveness of internet-based therapies for insomnia (*Lien et al., 2019*; *Ye et al., 2016*). As expected, compared with the IMBSR, the ICBT-I produced greater improvements in the severity of insomnia (*Cohen's d* $=1.37$) and depression (Cohen's $d = 0.71$) post-intervention. These findings add to the large body of research indicating that ICBT-I is an effective treatment for insomnia and depressive symptoms (*Brooks et al., 2018*; *Cheng et al., 2019*; *Shaffer et al., 2022*; *Zachariae et al., 2016*; *Turkowitch et al., 2022*; *Van Der Zweerde et al., 2019*).

The IMBSR significantly improved insomnia severity and depression. Research indicates that MBIs can enhance sleep quality in various clinical populations with sleep disturbances (*Adler et al., 2017*; *Garland et al., 2014*; *Gross et al., 2011*; *Ong et al., 2018*; *Peter et al., 2019*). In line with the findings of this study, therapist-guided online mindfulness interventions are effective in reducing insomnia and depressive symptoms (*Aslan, Aslan & Coşkun,*

*2021*; *Querstret, Cropley & Fife-Schaw, 2018*; *Kennett, Bei & Jackson, 2021*). Mindfulness techniques allow individuals to identify and effectively manage chronic or unproductive thought patterns and decrease heightened arousal before sleep (*Carletto et al., 2016*; *Kalmbach et al., 2023*). MBIs impact depression by reducing cognitive arousal, such as rumination and over-engagement, and increasing the capacity to tolerate negative emotions (*Altena et al., 2023*; *Mamede et al., 2022*).

Notably, both the IMBSR and ICBT-I improved mindfulness. This finding is consistent with previous studies of internet-delivered MBIs, which have shown improvements in mindfulness skills (*Sevilla-Llewellyn-Jones et al., 2018*; *Bossi et al., 2022*; *Jayewardene et al., 2016*). Surprisingly, the ICBT-I, without formal mindfulness training, revealed significant improvement in mindfulness. This observation indicates that the cognitive-behavioral aspects of ICBT-I indirectly contribute positively to mindfulness. This outcome could be attributed to the positive connection between trait mindfulness and quality of sleep (*Ding et al., 2020*). Additionally, cognitive restructuring addresses dysfunctional beliefs about sleep, selective attention, worry, and tracking automatic thoughts and may promote meta-awareness of thoughts and mindfulness abilities (*Altena et al., 2023*). CBT-I assists nurses in shifting metacognitions in a positive direction, addressing negative thoughts, and modifying erroneous beliefs about sleep, which can lead to improved sleep quality and a better understanding of sleep needs. In future CBT-I investigations, it would be advantageous to incorporate assessments of mindfulness to gain deeper insight into the underlying biological mechanisms through which CBT-I exerts its effects.

Group-based IMBSR and ICBT-I are feasible and acceptable for nurses with insomnia. Nurses demonstrated high satisfaction and adherence to both interventions, with 77.6% completing the study, demonstrating their commitment to research protocols. These findings align with the literature, suggesting that therapist-guided psychological interventions may enhance adherence outcomes (*Spijkerman, Pots & Bohlmeijer, 2016*). The results indicated that both programs were well tolerated. This study contributes to the growing body of evidence supporting the effectiveness of group-based internet-delivered interventions in promoting mental health among HCWs (*Luangphituck, Boonyamalik & Klainin-Yobas, 2023*). The high completion rate in this study suggests that group-based internet-delivered interventions are feasible and acceptable for nurses with insomnia and related mental health concerns. The group format may have contributed to adherence by providing a supportive and interactive environment. Additionally, the interventions were tailored to the specific needs of nurses and delivered by therapists, which may have further enhanced engagement and motivation. Nurses also reported high levels of satisfaction with both IDIs, highlighting the convenience, flexibility, and social support they offered.

### Limitations

The novel aspects of this study contribute to its clinical and theoretical significance. However, the study also has some limitations. One primary limitation of this trial was the absence of a placebo group. A placebo group would have allowed for a more robust evaluation of the specific effects of ICBT-I and IMBSR. The lack of objective and biological measures (*e.g.*, actigraphy or polysomnography) was another limitation of this trial.

Without follow-up data, we were unable to evaluate the long-term effects of the programs. Further studies with follow-up data are needed to determine the long-term durability of ICBT-I and IMBSR. The nurses were generally familiar with sleep hygiene and RCTs. Therefore, as well-educated participants, they may have been able to obtain more benefits from the delivered programs. Both programs were conducted *via* the internet. Therefore, the nurses were not limited by physical, geographical, or occupational problems, which increased the generalizability of the study. However, nurses without internet access were excluded from participation. Additionally, the online format may have made it more difficult for the therapists to monitor participants' progress. The researchers took into account a potential 25% dropout rate when determining the sample size, resulting in a sample of 134 participants. This careful consideration of attrition rates enhances the study's generalizability and its capacity to withstand participant attrition.

## CONCLUSIONS

Overall, the study suggested that both the ICBT-I and IMBSR are viable interventions for improving sleep quality in clinical nurses, with implications for clinical practice and research. ICBT-I stands out as a first-line treatment for nurses experiencing insomnia. This study supports the feasibility, acceptability, and tolerability of group-based IMBSR and ICBT-I, which contribute to the well-being of nurses and a resilient healthcare workforce. Comparative studies suggest that CBT-I may have a more immediate impact, while MBSR might offer long-term benefits in terms of sleep quality and overall well-being. Future research should explore a combined approach, merging CBT-I with mindfulness interventions for a more comprehensive insomnia treatment strategy. Internet-delivered treatments address geographical challenges but need further exploration of factors influencing participant adherence. Encouraging interdisciplinary collaboration can help address these gaps and provide innovative solutions. Qualitative studies can uncover experiences and improve interventions. Future trials should consider including a placebo group to enhance methodological rigor and gain nuanced insights into intervention effects, strengthening the study's design.

### Funding
Jiangxi University (Jiangxi Province Social Science Planning Youth Foundation Project (No. 21JY45) supported the APC. The funders had no role in study design, data collection and analysis, decision to publish, or preparation of the manuscript.

### Grant Disclosures
The following grant information was disclosed by the authors:
Jiangxi Province Social Science Planning Youth Foundation Project: No. 21JY45.

### Competing Interests
The authors declare there are no competing interests.

## Author Contributions

- Wanran Guo analyzed the data, authored or reviewed drafts of the article, interpretation and funding acquisition, and approved the final draft.
- Nabi Nazari conceived and designed the experiments, performed the experiments, analyzed the data, prepared figures and/or tables, authored or reviewed drafts of the article, and approved the final draft.
- Masoud Sadeghi performed the experiments, analyzed the data, authored or reviewed drafts of the article, and approved the final draft.

## Human Ethics

The following information was supplied relating to ethical approvals (i.e., approving body and any reference numbers):

The Lorestan Research Ethics Committee Lorestan University of Medical Sciences approved the study (IR.LUMS.REC.1399.269).

## Clinical Trial Ethics

The following information was supplied relating to ethical approvals (i.e., approving body and any reference numbers):

Lorestan University of Medical Sciences

## Ethics

The following information was supplied relating to ethical approvals (i.e., approving body and any reference numbers):

The Research Ethics Committees of Lorestan University of Medical Sciences granted approval (IR.LUMS.REC.1399.269).

## Data Availability

The raw measurements are available in the Supplementary File.

## Clinical Trial Registration

The following information was supplied regarding Clinical Trial registration:

ISRCTN36198096

## Supplemental Information

Supplemental information for this article can be found online at http://dx.doi.org/10.7717/peerj.17491#supplemental-information.

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
