# Peer review of "Cognitive-behavioral treatment for insomnia and mindfulness-based stress reduction in nurses with insomnia: a non-inferiority internet delivered randomized controlled trial"

_PeerJ, doi:10.7717/peerj.17491_

## Round 0.1 · original submission · Major Revisions

I have received the reports on your manuscript. Based on the advice received, your manuscript will need major revisions. Both reviewers agree that the work has the potential to provide an important contribution to the efficacy of “behavioral and cognitive” interventions for insomnia. When preparing your revised manuscript, you are asked to carefully consider the reviewers’ comments.

More specifically (but not exclusively), both reviewers suggest adding more details to the descriptions of the methodology adopted, as well as the presentation of statistical results. Related to this, reviewer 1 suggests deepening the statistical analysis.

·

Basic reporting

This original article aims at assessing the efficacy and the non-inferiority of a mindfulness-based program for insomnia (IMBSR) compared to an internet-delivered cognitive-behavioural therapy for insomnia (ICBT-I). At this purpose, 134 nurses with insomnia were randomly allocated to undergoing IMBSR or ICBT-I for eight weekly sessions of two hours. The pre- and post-intervention assessment comprised self-reported outcome measures, including the Insomnia Severity Index (ISI), Patient Health Questionnaire-9 (PHQ-9), 15-item Five Facet Mindfulness Questionnaire (FFMQ), and Client Satisfaction Questionnaire (CSQ-I). The main findings revealed that nurses in the ICBT-I group significantly improved insomnia and depressive symptoms compared to those in the IMBSR group. Moreover, nurses in the IMBSR group significantly improved mindfulness symptoms compared to those in the ICBT-I group. The experimental design and the methodology is sound, rigorous and the rationale in well explained.
The authors tackled a captivating and highly relevant topic with real-world clinical implications. Nevertheless, the manuscript has some aspects and therefore I would not recommend it for publication in its current form. Major revisions would improve the relevance and results of the project.

Major revisions:
Introduction: I would suggest authors to cite the following articles regarding the efficacy of the MBSR compared with nonspecific active controls (some studies included also CBT as a comparison term; doi: 10.1016/j.jpsychores.2020.110144; doi: 10.1111/nyas.13996).
Lines 65-66: “Defined as the nonjudgmental observation of the present moment, mindfulness encompasses both internal and external experiences”. Authors should stress the intentionality of paying attention, which is the base of the meditation and yoga, and in general of mindfulness.

Methods: sound, well structured, and explained. For transparency and reproducibility of science, authors already published a study protocol on the project (https://pubmed.ncbi.nlm.nih.gov/36527137/). Regarding intervention protocols, I would suggest authors to add some details in the MBSR protocol (already reported in Appendix) to improve the reproducibility of the intervention (i.e., minutes per meditation in each session; the materials used to inquiry the theoretical framework of each session (sleep, stress, etc); the materials used for homework; the potential utility of daily-life mindfulness such as mindful-sleep).

Statistical analysis: 1) Descriptive characteristics: Table 2. It is not clear what do authors mean with “value”, i.e. baseline value or IMBSR value. I suggest deleting value as the units of measures (percentages, means ± SD) are already specified in the table. Moreover, I suggest authors reporting the baseline parameters for marital status, age, insomnia, depression and mindfulness separated for each group, as well as in the manuscript text (lines 277-279). 2) Treatment results: the authors should perform both intention-to-treatment analysis, and as they stated in protocol study, a mixed effects model or repeated ANOVA with time (T1, T2) and group (IMBSR, ICBT-I) on the secondary outcomes. Moreover, I suggest authors considering p-value of analysis after Bonferroni’s correction for multiple comparisons. Last, in an exploratory-fashion, the percentages of nurses who respond to each intervention in terms of insomnia, depressive symptoms and mindfulness would be of help to interpret comprehensive findings of such trial.

Discussion: The authors should discuss potential similarities/differences in the methodology, participants, procedures, treatment of data with previous studies, which found positive results for insomnia. The practical implications should be stressed.

Experimental design

Research questions are well-defined, the investigation was conducted rigorously and to a high technical standard. Methods were clear and with sufficient details. Refer to Basic Reporting for revisions.

Validity of the findings

The validity of the findings should be improved as suggested in the Basic Reporting. The impact of MBSR-I and the ICBT-I in improving insomnia, depressive symptoms and mindfulness need properly statistical analysis, in order to support such findings. The comprehensive report of results in a figure would be of help.

Additional comments

English proofreading would be necessary.

Reviewer 2 ·

Basic reporting

1. The introduction is adequate and the authors adequately justify the basis of their treatments. In line 49, I suggest adding a brief description of the factors that Cognitive-Behavioral Therapy for Insomnia (CBT-I) addresses, such as sleep drive dysregulation, behaviors and cognitions interfering with sleep, attempts to control the sleep process, and sleep-related anxiety (Baglioni et al., 2020). Please insert: Baglioni et al. (2020). The European Academy for Cognitive Behavioural Therapy for Insomnia: An initiative of the European Insomnia Network to promote implementation and dissemination of treatment. Journal of Sleep Research, 29(2), Article e12967
2. Regarding the introduction of MBI (line 64), I suggest mentioning that Segal et al. (2002) developed Mindfulness-Based Cognitive Therapy (MBCT) with the intention of using mindfulness principles to disrupt negative ruminative thought patterns that contribute to depression relapse (Segal, Williams, & Teasdale, 2002). This provides an important theoretical basis for understanding the application of mindfulness in different contexts. Segal, Z. V., Williams, J. M. G., & Teasdale, J. D. (2002). Mindfulness-based cognitive therapy for depression: A new approach to preventing relapse. The Guilford Press.
3. When the authors indicate that "Mindfulness-based interventions (MBIs) complement CBT-I by integrating mindfulness techniques" (line 73), I suggest providing a more specific description of how mindfulness techniques are integrated with CBT-I to enhance readers' understanding of this integration.
4. In line 82, where it is mentioned that "Mindfulness-based stress reduction (MBSR) has emerged as a promising approach for tackling insomnia and sleep disturbances," I suggest adding the reference from Camino et al. (2022) to support this claim and provide additional evidence on the effectiveness of this intervention in reducing the severity of insomnia symptoms. Please inserte: Camino, M., Satorres, E., Delhom, I., Real, E., Abella, M., & Meléndez, J. C. (2022). Mindfulness-based cognitive therapy to improve sleep quality in older adults with insomnia. Psychosocial Intervention, 31(3), 159-167.
5. Regarding lines 106 and 110, where the phrase is repeated, I suggest removing one of the repetitions to avoid redundancies.

6. Regarding Figure 1, I suggest improving it by including a clearer representation of the study phases, such as pre-treatment, post-treatment, and dropout, as well as clearly indicating the number of participants assigned to each group in each phase. Additionally, the abbreviations in the caption should match the elements in the figure to avoid confusion.

Experimental design

1. The research design is appropriate, and the research question is relevant to the insomnia literature. However, there's a point that could be clarified. In line 133, when you mention “24 who did not meet the insomnia severity cutoff score”, it would be helpful to specify the cutoff point. This aspect raises a question for me. You are using the ISI, and indeed the cutoff point is 15. However, this scale distinguishes between moderate insomnia (15-21) and severe insomnia (22-28). I understand that you are working with both groups, but it would be beneficial to clearly indicate this in your inclusion criteria, instead of simply saying that you include ≥14. Additionally, both groups have different clinical characteristics, so it would be interesting to differentiate these subgroups in the analyses. The evolution of the groups may be different. I recognize that this may pose a challenge due to sample size, but if so, you should clearly inform and justify why you are working with both groups.
2. The results are statistically robust. As a minor suggestion, you should include the cutoff points for partial effect sizes. Additionally, it would be beneficial to include some comments on effect sizes in the discussion section. This could strengthen your arguments when comparing interventions.

Validity of the findings

1. The research and its conclusions are interesting. Comparing two treatments is a suitable strategy to strengthen the validity of these findings. Additionally, the study is replicable. The target population for the treatments is appropriate, as they are under stress due to their work, and non-pharmacological treatments like these could be a good alternative. The conclusions are well-formulated and provide valuable information to the field of study.

Additional comments

no comment'

---

## Round 0.2 · accepted · Accept

Dear authors, I would like to thank you for having addressed the reviewers' comments.

In my opinion, the manuscript is ready for publication.

Best regards

·

Basic reporting

Thanks to the authors for the revisions.
The manuscript still needs improvement.

Statistical analysis paragraph:
Thanks to the authors for having been added this part of the percentages of nurses who respond to each intervention in terms of insomnia, depressive symptoms and mindfulness would be of help to interpret comprehensive findings of such trial. I suggest to add this part also in the statistical analysis paragraph. It would aid the readers.

Experimental design

Thanks to the authors for the revisions.
See Basic Reporting.

Validity of the findings

Thanks to the authors for the revisions.
See Basic Reporting.

Additional comments

Thanks to the authors for the revisions.
See Basic Reporting.

Reviewer 2 ·

Basic reporting

The changes made have resolved the suggestions.

Experimental design

The changes made have resolved the suggestions.

Validity of the findings

The changes made have resolved the suggestions.

Additional comments

no coments